# The Psychoemotional Stress-Induced Changes in the Abundance of SatIII (1q12) and Telomere Repeats, but Not Ribosomal DNA, in Human Leukocytes

**DOI:** 10.3390/genes13020343

**Published:** 2022-02-14

**Authors:** Pavel E. Umriukhin, Elizaveta S. Ershova, Anton D. Filev, Oksana N. Agafonova, Andrey V. Martynov, Natalia V. Zakharova, Roman V. Veiko, Lev N. Porokhovnik, George P. Kostyuk, Sergey I. Kutsev, Natalia N. Veiko, Svetlana V. Kostyuk

**Affiliations:** 1Research Centre for Medical Genetics, Laboratory of Molecular Biology, 115522 Moscow, Russia; pavelum@mail.ru (P.E.U.); es-ershova@rambler.ru (E.S.E.); afilev92@gmail.com (A.D.F.); agafonova-o@ukr.net (O.N.A.); avlamar@mail.ru (A.V.M.); veikor@rambler.ru (R.V.V.); kutsev@mail.ru (S.I.K.); satelit32006@yandex.ru (N.N.V.); svet-vk@yandex.ru (S.V.K.); 2Faculty of Normal Physiology, I.M. Sechenov First Moscow State Medical University, 119991 Moscow, Russia; 3P.K. Anokhin Institute of Normal Physiology, 125009 Moscow, Russia; 4Federal Research and Clinical Center of Intensive Care Medicine and Rehabilitology, V.A. Negovsky Research Institute of General Reanimatology, 107031 Moscow, Russia; 5Moscow Healthcare Department, N.A. Alexeev Clinical Psychiatric Hospital #1, 117152 Moscow, Russia; nataliza80@gmail.com (N.V.Z.); kgr@yandex.ru (G.P.K.)

**Keywords:** CNVs, telomere, satellite DNA, schizophrenia, stress

## Abstract

INTRODUCTION. As shown earlier, copy number variations (CNV) in the human satellite III (1q12) fragment (f-SatIII) and the telomere repeat (TR) reflects the cell’s response to oxidative stress. The contents of f-SatIII and TR in schizophrenic (SZ) patients were found to be lower than in healthy controls (HC) in previous studies. The major question of this study was: ‘What are the f-SatIII and TR CNV dynamic changes in human leukocytes, depending on psychoemotional stress?’ MATERIALS AND METHODS. We chose a model of psychoemotional stress experienced by second-year medical students during their exams. Blood samples were taken in stressful conditions (exams) and in a control non-stressful period. Biotinylated probes were used for f-SatIII, rDNA, and TR quantitation in leukocyte DNA by non-radioactive quantitative hybridization in SZ patients (*n* = 97), HC (*n* = 97), and medical students (*n* = 17, *n* = 42). A flow cytometry analysis was used for the oxidative stress marker (NOX4, 8-oxodG, and γH2AX) detection in the lymphocytes of the three groups. RESULTS. Oxidative stress markers increased significantly in the students’ lymphocytes during psychoemotional stress. The TR and f-SatIII, but not the rDNA, contents significantly changed in the DNA isolated from human blood leukocytes. After a restoration period (post-examinational vacations), the f-SatIII content decreased, and the TR content increased. Changes in the blood cells of students during examinational stress were similar to those in SZ patients during an exacerbation of the disease. CONCLUSIONS. Psychoemotional stress in students during exams triggers a universal mechanism of oxidative stress. The oxidative stress causes significant changes in the f-SatIII and TR contents, while the ribosomal repeat content remains stable. A hypothesis is proposed to explain the quantitative polymorphisms of f-SatIII and TR contents under transient (e.g., students’ exams) or chronic (in SZ patients) stress. The changes in the f-SatIII and TR copy numbers are non-specific events, irrespective of the source of stress. Thus, our findings suggest that the psychoemotional stress, common in SZ patients and healthy students during exams, but not in a schizophrenia-specific event, was responsible for the changes in the repeat contents that we observed earlier in SZ patients.

## 1. Introduction

Copy number variations (CNVs) of certain genome fragments are typical for the human genome [1,2]. CNVs are considered to determine predispositions to a number of diseases [2,3,4,5,6,7,8]. Tandem repeats in the human genome are characterized by enhanced instability with a pronounced quantitative polymorphism [7,9,10,11]. Previously, we studied the CNVs of three tandem repeats: ribosomal (rDNA), satellite III, and telomere repeats [12,13,14,15,16,17,18,19].

**Ribosomal repeats (rDNA)**: The rDNA is located on acrocentric chromosomes. Each rDNA unit consists of a transcribed region that includes three rRNA genes (18S, 5.8S, and 28S) and a non-transcribed spacer. In the nucleus, rDNA forms the nucleolus, a special structure where rDNA transcription occurs, and the initial events of ribosome biogenesis occur [20,21,22,23,24]. The rDNA copy number (CN) in different human genomes varies in a wide range from 200 to 900 copies [14,19]. We previously found [12,19] that genomes of schizophrenia patients (SZ) contained increased rDNA CNs compared to healthy controls (HC).

Our earlier data suggest that the rDNA CN index is a stable genetic trait that does not change in response to stress and aging. We did not find rDNA content heterogeneity in different cells of the same person, and we observed no effects of genotoxic stress on rDNA CNs in human cells [14,16,18]. However, the rDNA content can vary significantly in cancer cells, compared to the non-cancerous cells of the same individual [25].

**Satellite III repeat (f-SatIII):** A 1.77-kb fragment from satellite III [26] is a part of the largest pericentromeric heterochromatin of the first chromosome (1q12). The human genome contains from 5 to 45 pg of the repeat per one ng of total DNA. The f-SatIII content in DNA samples isolated from children was significantly reduced compared to the samples isolated from adults [15]. We also observed significantly reduced repeat copy numbers in the genomes of SZ patients compared to the control cohort [16]. The range of the f-SatIII content variation in DNA was significantly increased in the senile sample and in the sample of nuclear industry workers. During the replicative aging of human skin fibroblasts (HSFs), or in response to genotoxic stress, such as ionizing radiation or Cr(VI) salts, we observed significant changes to the f-SatIII content in cellular DNA [15,18]. The HSF populations were heterogeneous to the f-SatIII content. During early passages, we found a three-fold f-SatIII content variation. In DNA isolated from various brain regions of a SZ patient, we also found significant f-SatIII content variability [15,16]. The f-SatIII content varies significantly in cancer cell pools and may increase or decrease in tumor cells [27]. We concluded that, unlike rDNA CNs, the f-SatIII repeat content in the human genome is not a stable genetic trait and may significantly change with aging and oxidative stress.

**Telomere repeat (TR):** Repeat (TTAGGG)_n_ is localized at the ends of chromosomes in the telomere regions. The telomere sizes may differ between different arms of the same chromosome, as well as homologous and non-homologous chromosomes, different cells, tissues, organs, between twins, different individuals belonging to the same species, and different species. The nature and functional significance of this variability are still not clear [28,29,30]. The literature presents ambiguous data on the TR content in the genomes of SZ patients [31,32]. The TR content in the human body is determined by many factors. For example, some results suggest that chronic stress is associated with shorter telomeres in human PBMCs [33]. Oxidative stress, mutations in DNA repair genes, and epimutations may be among the major mechanisms of telomere attrition [34,35]. During the replicative aging of HSFs, we observed a negative correlation between TR and f-SatIII contents. A similar relationship was found when we studied the contents of the three repeats (rDNA, f-SatIII, and TR) in DNA isolated from the various brain regions of an SZ patient [15,16]. Thus, the TR content in human DNA is not a stable genetic trait, similar to f-SatIII repeats.

A disadvantage of our CNV studies of the three repeats is that almost all the data were obtained on different subjects or in vitro. We did not conduct in vivo studies demonstrating the changes in the abundance of the three repeats in the same subject under stress or disease. Such an experiment should include the analysis of the same person’s DNA during various periods of life. The non-specific stress reaction can be induced by aging, disease, or activities involving significant mental, emotional, or physical loads over a certain period. From all these options, we selected a model of psychoemotional stress experienced by medical students during exams for the present study. The model is often used by researchers and has been well studied [36,37,38,39,40,41,42,43,44,45,46,47]. In addition, this model helps us to better understand the tandem repeat content changes in the genomes of SZ patients, who also experience chronic stress, especially during the periods of exacerbation and hospitalization [48].

In the present study, we have shown that psychoemotional stress induces significant variations in the contents of f-SatIII and TR in medical students during exams, whereas the rDNA content remains stable in the leucocyte DNA. Previously, we suggested that oxidative stress was one of the main conditions that causes changes in f-SatIII content [18]. To assess the level of oxidative stress in blood cells, we measured NOX4 and γ-H2AX proteins, and a DNA oxidation marker (8-oxodG), in blood lymphocytes in the three groups studied.

## 2. Materials and Methods

### 2.1. Population Samples

The dataset consisted of 211 subjects inhabiting Moscow city and the surrounding suburbs. The subjects were divided into three groups. The HC-group (*n* = 97) included healthy subjects. The SZ-group (*n* = 97) included drug-naïve SZ patients with a severe psychotic episode that required hospitalization. The clinical and demographic characteristics of the HC- and SZ-groups are shown in Table 1.

The St-group included 19–20 year-old second-year medical students (*n* = 17, *n* = 42). These students passed three exams during the exam session. Blood sampling was performed before the exams, on the day of the second exam, and after the university vacations. The St(I) group included 17 students during the preparation for the exam (30 days before the second exam). That period is considered the initial period of stress. Students had a pronounced mental load associated with writing test papers, passing preliminary tests, and preparing for the exam session. The St(II) group included 15 students out of 17 who agreed to continue their participation in the study. The blood samples were taken directly at the day of the exam (9 a.m.) two hours before the second exam (physiology examination). This period is considered a period of intense emotional and mental stress. The St(III) group included 10 of the 15 St(II) group’s students who were available for the final stage. The blood samples were collected 30 days after the second exam (20 days after the end of the exam session). The students had restored during a two-week vacation and the residual stress associated with the exams was minimal.

#### 2.1.1. The SZ Group

Blood samples were obtained from N. A. Alexeev Clinical Psychiatric Hospital #1 SZ patients. Patients were diagnosed with paranoid SZ (F20.00 or F20.01) according to the International Classification of Diseases 10 criteria, using structured interviews (Mini-International Neuropsychiatric Interview). Diagnoses were also confirmed pursuant to the Diagnostic and Statistical Manual of Mental Disorders, 4th Edition criteria. The SZ group included drug-naïve patients in their first episode and patients with a chronic form who had not taken antipsychotics at least 2–4 weeks before hospitalization.

#### 2.1.2. The HC Group

Peripheral blood was collected from the HC group in the Research Centre for Medical Genetics. The HC group included volunteers (men and women, Table 1) aged 17 to 51. All the participants had no pronounced hereditary diseases, were mentally healthy, and had never sought psychiatric help. This group included employees of the institutions that conducted this study and the healthy relatives of patients with non-psychiatric diagnoses observed at the Research Center for Medical Genetics.

#### 2.1.3. The St Group

Second-year medical students (*n* = 17, *n* = 42) of the I.M. Sechenov First Moscow State Medical University were included in the study (11 male and 6 female subjects, 19–20 years old). Subjects had to be healthy for their inclusion in the study. The exclusion criteria were stress diagnoses, with medication-influencing biomarkers of stress. Blood samples were taken during a control non-stressful period (30 days after an examination and a vacation, group St(III), *n* = 10) and in stressful conditions on the day of the examination (9 am) two hours before a physiology exam (group St(II), *n* = 15) and 30 days before the exam (group St(I), *n* = 17). Each study participant gave written informed consent.

#### 2.1.4. The Patient’s Consents to the Analyses Performed

The study was carried out in accordance with the latest version of the Declaration of Helsinki and was approved by the Independent Interdisciplinary Ethics Committee of Ethical Review for Clinical Studies (51 Leningradsky Prospekt, Moscow, 125468, Russia, +7-915-346-30-30), Protocol #4, as of 15 March 2019, for the scientific minimally interventional study ‘Molecular and neurophysiological markers of endogenous human psychoses’. Each participant signed informed written consent to participate in the study after the procedures had been completely explained.

### 2.2. DNA Isolation from Leukocytes

Five milliliters of blood was collected from the peripheral vein using a syringe flushed with heparin (0.1 mL/5 mL blood) under strict aseptic conditions. The leukocytes were isolated from 5 mL of blood according to Boyum’s technique [49]. To isolate the DNA, we used the standard method described in detail previously [19]. Briefly, 5 mL of the solution (2% sodium lauryl sarcosylate, 0.04 M EDTA, and 150 μg/mL RNAse A (Sigma, St. Louis, MO, USA)) was added to the fresh leucocytes for 45 min (37 °C) and then treated with proteinase K (200 μg/mL, Promega, Madison, USA) for 24 h at 37 °C. The lysate samples were extracted with an equal volume of phenol, phenol/chloroform/isoamyl alcohol (25:24:1), and chloroform/isoamyl alcohol (24:1), respectively. DNA was precipitated by adding 1/10 of a volume of 3 M sodium acetate (pH 5.2) and a 2.5 volume of ice-cold ethanol. DNA was collected by centrifugation (10,000× *g* for 15 min at 4 °C), washed with 70% ethanol (*v*/*v*), and dissolved in water. The determination of the DNA concentration was carried out in two steps. The first one gives a rough estimate of the initial amount of DNA in each sample using UV spectroscopy. At the end of the first step, the amount of DNA needed to make a 60 ng/μL solution of DNA was calculated. The final quantification uses this solution to calculate the exact amount of DNA needed to dilute samples to 50 ng/μL. The final DNA quantification was performed fluorimetrically using a PicoGreen dsDNA quantification reagent by Molecular Probes (Invitrogen, Carlsbad, CA, USA). The assay displays a linear correlation between the dsDNA quantity and the fluorescence within a wide range of concentrations. The DNA concentration in the sample is calculated according to a DNA standard curve. We used EnSpire equipment (Finland) at λ_ex_= 488 nm and λ_em_= 528 nm.

### 2.3. Nonradioactive Quantitative Hybridization (NQH)

The NQH technique is based on complementary hybridization of the studied DNA, immobilized on a filter, with a fluorescently-labeled DNA-probe. The standard samples of the genomic DNA, with a known content of the repeat, were applied to the same filter in order to plot a signal intensity-repeat copy number calibration curve. After hybridization, the filter was scanned, and the dot signal intensity was measured using special software.

The NQH was used for the quantification of f-SatIII, TR, and rDNA repeats, as specified in detail previously (Supplements of [15,17] and reference [19]). We made no modifications in the technique described. For calibration, we used five standard human DNA samples with known f-SatIII, TR, and rDNA copy numbers. The relative standard error for NQH was merely 5 ± 2%. The major overall error of the experiment was contributed by the step of isolating DNA from leukocytes. The total standard error was 11 ± 7%.

Below, the DNA probes are detailed.

#### 2.3.1. DNA Probe for SatIII Quantification (f-SatIII)

The f-SatIII probe was a 1.77-kb cloned EcoRI fragment of human satellite DNA [26] labeled with biotin-11-dUTP using nick translation. Dr. H. Cook (MRC, Edinburgh, UK) kindly supplied the human chromosome lql2-specific repetitive satellite DNA probe pUC1.77.

#### 2.3.2. DNA Probe for rDNA Quantification

The human rDNA pBR322-rDNA probe contains rDNA sequences (5836 bp) cloned into an EcoRI site of the pBR322 vector. The rDNA fragment clones covered positions from −515 to 5321 of the human rDNA (GenBank accession No. U13369).

#### 2.3.3. DNA Probe for Telomere Repeat Quantification

For the detection of the human telomere repeat, the following probe was used: biotin-(TTAGGG)_7_. Syntol (Moscow, Russia) performed the synthesis and biotin labeling of the oligo-probe.

#### 2.3.4. Standard Calibration Curve for TR Quantification

Six standard samples of genomic DNA, with a known TR content, were applied to the same filter in order to plot a calibration curve for the dependence of the signal intensity on the TR content in a particular sample.

To obtain the calibration samples with the known TR content, we added the model TR fragment (1.1 kb) to the sample of human DNA with the lowest TR content, with an amount of 0.1 to 1 picograms per ng (100 to 1000 pg/μg) of the genomic DNA. Using this procedure, six standard samples of genomic DNA with various TR contents were selected.

The dependence of the TR content on the relative hybridization signal (Figure 1) is linear and is well-reproduced in independent experiments. Therefore, for each specific experiment, two calibration samples could be used further, instead of six.

### 2.4. Flow Cytometry Assay (FCA)

Lymphocytes were isolated from 15 mL of blood. Then, the lymphocytes were separated by centrifuging at 1500 rpm for 30 min using lymphocyte separation media (Histopaque 1077 with a density of 1.077 g/mL, Sigma). Cells were washed with PBS, then centrifuged, and resuspended in PBS. To fix the cells, the paraformaldehyde (Sigma) was added at a final concentration of 2% at 37 °C for 10 min. Cells were washed three times with 0.5% BSA–PBS and were permeabilized with 0.1% Triton X-100 (Sigma) in PBS for 15 min at room temperature.

Cells (5 × 10^4^) were washed three times with 0.5% BSA–PBS and were stained with 1 µg/mL of the FITC–(NOX4, 8-oxodG, γH2AX, NF-kB, p53, MDM2, NRF2, BCL2, BAX1, LC3, MRE11A, RAD50, and BRCA1) antibody (Abcam) for 4 h at 4 °C in the dark, then were again washed thrice with 0.5% BSA–PBS. To quantify the background fluorescence, we stained a portion of the cells with secondary FITC-conjugated antibodies. The cells were analyzed at CytoFLEX S (Beckman Coulter). The area of T-lymphocytes (CD3+) was determined on an SSC-FCS plot (see the Appendix A). Primary data are presented as the median values of the signal. In each experiment, for comparison, we analyzed a control standard sample of lymphocytes from a healthy donor. The index FL1-Xi (arbitrary units) is a ratio of the median values of the experimental and control signals. The relative standard error of the FL1-Xi index was 4 ± 2%.

### 2.5. Statistical Analyses

All the findings reported here were reproduced at least two times as independent biological replicates. The significance of the observed differences was analyzed using a non-parametric Mann–Whitney U test and Kolmogorov–Smirnov statistics. A spearman analysis and a linear regression analysis were used to analyze correlations between the two parameters. Data were analyzed with StatPlus2007 professional software. Available online: http://www.analystsoft.com (accessed on 10 January 2022). All *p*-values were two-sided and were considered statistically significant at *p* < 0.05.

## 3. Results

We analyzed the contents of f-SatIII, TR, and rDNA in blood leukocyte DNA in 211 individuals divided into three cohorts, as described in Section 2 (the 97 HC group, 97 SZ group, and 17 St group).

### 3.1. rDNA, f-SatIII, and TR CNVs under Stress

**rDNA CN:**Figure 2A1–A3,D shows the rDNA contents in leucocytes from the studied groups. SZ genomes contained more rDNA copies than the HC or St genomes (Figure 2A and Table 2 and Table 3). Thus, we confirmed the results previously found in other samples of SZ patients and in healthy controls [12,19]. The rDNA CN in leucocytes of 17 students varied in a very narrow range (C*v* = 0.12) compared to HC (C*v* = 0.16) and SZ (C*v* = 0.22) samples. The rDNA CN variability index in the two (St I–II, *n* = 15) or three (St I–III, *n* = 10) DNA samples of the same students was less than 10% and was comparable to the relative standard error of the experiment (Figure 2A3). Thus, the rDNA content in the students’ genomes was a stable genetic trait that did not depend on psychoemotional stress.

**f-SatIII**: The f-SatIII content in the leucocytes from SZ patients was reduced compared to HC group (Figure 2B1, Table 2 and Table 3). We corroborated the result previously found in other samples of SZ patients and healthy controls [16]. In contrast to rDNA, the leucocyte content of f-SatIII in student DNA significantly changes under stress (Figure 2B). Most students showed that, at the day of the exam and especially after their vacations, the f-SatIII content was significantly lower than the content determined 30 days before the exam (Figure 2B2,B3). In two students (the curves are indicated as M and *n*), the f-SatIII DNA amount was at its maximum on the day of the exam, and then decreased in the same way, as shown by the others, remaining elevated in comparison with the other study participants.

**TR:** The TR content in the leucocyte DNA varied by a much broader range (C*v* (SZ) = 0.70; C*v* (HC) = 0.58) than f-SatIII (C*v* (SZ) = 0.26; C*v* (HC) = 0.31) and, especially, rDNA (C*v* (SZ) = 0.22; C*v* (HC) = 0.16). The TR content in the DNA samples from SZ patients was reduced in comparison with the controls (Figure 2C1,D, Table 2 and Table 3). The TR content in the students changed significantly under stress. In the St(I) and St(II) subgroups, the TR copy numbers were less than in the St(III) subgroup (Figure 2C1). The analysis of the TR changes in each student showed that, on the day of the exam, and especially after the vacations, the repeat content significantly increased in most subjects (Figure 2C2,C3). Only two students (M and N) experienced a significant f-SatIII jump on the exam day, where TR decreased on the exam day and remained relatively low after the vacations, compared to 30 days before the exam.

Therefore, TR and f-SatIII contents are not stable genetic traits and can alter under the psychoemotional stress.

### 3.2. Oxidative Stress Markers in Lymphocytes from the Different Groups Studied

We analyzed oxidative stress markers using a flow cytometry assay (see the Appendix A). We found that some changes in the repeat contents detected during the examinational period correlated with the changes observed during schizophrenia exacerbation.

**NADPH-oxidase 4 (NOX4):** NOX4 is found in the cell membrane, and mitochondrial oxidoreductase catalyzes the hydrogen peroxide formation. Presumably, NOX4 is the main oxidative stress factor in a number of diseases [50]. The NOX4 protein expression in the lymphocytes of SZ patients was significantly higher than in HC and St groups. The NOX4 level in the St group under stress occupied an intermediate position between the HC and SZ groups (Figure 3A1 and Table 4). A combined analysis of the three student sub-cohorts did not reveal any significant differences in the levels of NOX4 (Table 3).

Figure 3A2 and Figure 4 (NOX4 graph) display NOX4 changes in each student’s lymphocytes depending on the blood sampling moment of life. In Figure 3A2 the starting points reflect NOX4 levels 30 days before the exam (group St(I)). On the day of the exam, NOX4 levels increased in six students (*p* < 0.05) and decreased in seven students (*p* < 0.05). In two students, NOX4 level did not change (*p* > 0.05). After the vacations, NOX4 levels dropped in most students. A particularly strong post-vacational decrease was observed in comparison with the St(I) point. The variation coefficient of the parameter was significantly reduced in group St(III) compared to the other two groups (Table 4).

In the lymphocytes of cases M and N, abnormal f-SatIII and TR changes correlated with the NOX4 maximum level during the period of stress (Figure 4, graph NOX4, points StI and/or StII).

**8-oxo-2’-deoxyguanosine (8-oxodG):** 8-oxodG is an oxidized derivative of deoxyguanosine. 8-oxodG is the major product of DNA oxidation. The concentration of 8-oxodG in the cell reflects the level of oxidative stress [51,52]. Figure 3A3 and Figure 4 (8-oxodG graph) and Table 4 show the 8-oxodG levels in the three groups studied. In Table 3, comparisons were made using non-parametric statistics and a ROC analysis. The DNA was more oxidized in SZ patients than in the HC group (Table 3).

The lymphocyte DNA oxidation changes for each separate student are shown in Figure 3A4. On the day of the examination, the DNA oxidation marker increased in four cases and decreased in three student cases. After their vacations, the level of DNA oxidation was low in each student, and the variation coefficient decreased significantly (Table 4). In the lymphocytes of cases M and N, we observed the maximum 8-oxodG level within the stressful period (Figure 4, graph 8-oxodG, points I and/or II)).

**Phosphorylated serine 139 histone H2AX (γ-H2AX):** γ-H2AX is normally used as a biomarker of the cellular response to DNA damage in human subjects. An assay for γH2AX generally reflects the presence of double-strand DNA breaks [53]. Figure 3A5 shows that γ-H2AX levels in the lymphocytes of HC and SZ groups did not differ significantly.

In the student groups, the γ-H2AX amount increased significantly under stress 30 days before the exam. However, on the day of the exam, γ-H2AX level decreased in most students (Figure 3A6 and Figure 4 (γ-H2AX graph) and remained low after their vacations. During the stress period (points I and II), the γ-H2AX level was at its maximum in case N (Figure 4).

Figure 3B1 shows a γ-H2AX correlation with 8-oxodG in the studied groups. In all the groups, we found a positive correlation between the two parameters, reflecting the level of lymphocyte DNA damage. Figure 3B2,B3 illustrates the dependence of the markers of DNA damage on the NOX4 level. A positive correlation was only found in the students under examination stress and after their vacations. High NOX4 values correlated with high DNA damage levels.

Thus, during the stress period, some students had significantly enhanced lymphocyte DNA damage against relatively high NOX4 expression. After the vacations, the DNA damage in the students decreased significantly. Among the students, cases M and N were particularly distinguished by pronounced DNA damage against high NOX4 protein concentrations.

### 3.3. Relationships between rDNA CNs and Oxidative Stress Markers

We have shown that rDNA CNs are stable genetic markers that do not depend on the oxidative stress in the cells. However, rDNA CNs in the cells can influence the parameters of oxidative stress [54]. Therefore, we determined how rDNA CNs could modulate the levels of oxidative stress markers.

Figure 5A (graphs 1, 4, and 7) shows the dependence of NOX4, 8-oxodG, and γ-H2AX levels on rDNA CNs in the genomes of the HC, SZ, and St (I–III, *n* = 17, *n* = 42) groups. Studying the graphs (1 and 4) did not reveal any dependence of NOX4 or γ-H2AX on rDNA CNs for the three groups. The FL1-8-oxodG index negatively correlated with rDNA CNs in the SZ group (Figure 5A4). Next, we split the rDNA values into subgroups (Figure 5B1,B4,B7).

***Low rDNA CN (<350):*** In the HC and SZ samples, the FL1-NOX4 and FL1-8-oxodG medians reached the maximum values among all the subgroups. The NOX4 and 8-oxodG levels in the cells of SZ patients were higher than in the controls. These subgroups did not differ significantly in the levels of γ-H2AX, a marker of double-strand DNA breaks.

***Medium rDNA CN (350 to 500)***: In this subgroup, the rDNA CNs were higher, and the mean NOX4 level decreased, as compared to the low-copied cases. The NOX4 and 8-oxodG levels in the SZ patients of that subgroup were higher than in the controls. These subgroups did not differ significantly in the levels of γ-H2AX.

***Large rDNA CN (>500)***: The largest rDNA CNs in these subgroups were associated with the lowest NOX4 and 8-oxodG levels. These stress indicators were especially reduced in the DNA of SZ patients with very high rDNA copy numbers (>680 copies), corroborating the findings of our earlier experiments [55].

Thus, our findings suggest that low rDNA CNs are associated with higher levels of oxidative stress and DNA oxidation, and vice versa (Figure 5D).

### 3.4. Relationship between Oxidative Stress Markers and Abundance of f-SatIII and TR

As we hypothesized, the TR and f-SatIII contents are not stable genetic traits, as they change in response to stress. Apparently, a quantitative polymorphism of these repeats is determined by the intensity of oxidative stress affecting the cell’s genome. To test this assumption, we analyzed an association between the f-SatIII and TR contents and the levels of oxidative stress markers.

***f-SatIII:*** Figure 5A2,A5,A8 and Figure 5B2,B5,B8 show the leucocyte f-SatIII content in the tested subjects, depending on the lymphocyte NOX4, 8-oxodG, and γH2AX levels in the same subject. Figure 5C summarizes all the data on a single graph (brown lines) for clarity. The data are normalized by the f-SatIII values in the relevant HC subgroups, with low levels of stress markers.

For the HC group, the (FL1-NOX4 or FL1-8-oxodG)–f-SatIII relations are better described by second-degree equations (Figure 5A2,A5). The f-SatIII content in the DNA of the HC subgroups decreased with increasing FL1-γH2AX markers.

For the SZ subgroups with different stress marker levels, we observed different relationships (Figure 5C). The f-SatIII content in the SZ-group decreased with increasing levels of all markers (NOX4, 8-oxodG, and γH2AX). The dependence was most pronounced for the DNA damage markers, such as 8-oxodG and γH2AX. The analysis showed that in the SZ subgroup with very low f-SatIII index values (<8 pg/ng, *n* = 7), which had not been found in the HC sample, the highest levels of oxidative stress markers NOX4, 8-oxodG, and γ-H2AX were observed. In other words, a very low f-SatIII content in the DNA of the SZ patients may indicate a high level of oxidative stress in the patient’s cells. Higher f-SatIII contents in the DNA of SZ patients is associated with lower DNA damage levels.

***TR***: Figure 5A3,A6,A9, Figure 5B3,B6,B9 and Figure 5C (blue lines) show a link between the TR index and the level of oxidative stress markers in lymphocytes. The maximum values of the TR index were observed in the HC subgroups with medium and high 8-oxodG levels. The TR content was reduced in the subgroup with high NOX4 levels. In the HC subgroup with an average γH2AX level, the TR content was reduced most pronouncedly.

In the SZ group, the maximum TR levels were observed at low 8-oxodG and γH2AX values. An increase in 8-oxodG and γH2AX correlated with a decrease in the TR content. The smallest TR numbers were found in the SZ subgroup with high levels of γH2AX. The TR content was weakly correlated with NOX4.

### 3.5. Changes in the Levels of Proteins Involved in the Exam Stress Response in Students

We observed an increase in the three transcription factors in students’ lymphocytes in response to stress: NF-kB, p53 (and its regulator, MDM2), and NRF2 (Figure 4). Cases M and N, who demonstrated the maximum stress markers during stressful periods I and II, also showed higher levels of the three factors under stress. During preparation and on the day of the exam, the levels of anti-apoptotic BCL2 and pro-apoptotic BAX1 proteins, the autophagy marker LC3, and the amounts of proteins involved in DNA repair (MRE11A, RAD50, and BRCA1) increased in the students’ lymphocytes.

## 4. Discussion

We applied a non-radioactive quantitative hybridization (NQH) technique, which has been specially developed for the quantification of tandem repeats that cannot be quantified correctly using quantitative PCR. The NQH method itself, its applicability to the CNVs analysis of tandem repeats, and its disadvantages, have been described in detail in several publications [17,19].

In this study, we analyzed the contents of rDNA, f-SatIII, and TR in the DNA of the medical students (19–20 years old), SZ patients, and HC. In SZ patients, the abundance of the repeats was quantified during the acute psychosis exacerbation, when emergent hospitalization was required. Thus, we studied two models of psychoemotional stress: a transient stress in young and healthy students caused by external impacts (a hard educational process), and an exacerbation of chronic stress caused by the disease. It should be noted that in some SZ patients, the psychosis exacerbation was also associated with traumatic emotional influences. The SZ etiology is known to include both genetic predispositions and traumatic environmental factors [56].

This study was based on our earlier studies on f-SatIII, TR, and rDNA CNVs in SZ and aging [14,15,16,19]. Those studies have shown the stability of individual rDNA CNs and quantitative polymorphisms of individual TR and f-SatIII abundance in the genome. The combined CNV analysis of the three repeats in the same subject during different periods of life was not conducted in the previous studies. We performed such an analysis only for SZ patients [16], comparing the f-SatIII content in their DNA on the day of hospitalization and after a month-long course of antipsychotic therapy. We found a significant variability in the contents of the f-SatIII repeat in the same patient, but it was not clear whether the variability was associated with the biochemical effect of the antipsychotics or with the stress alleviation due to the drug and psychotherapy. A study of medical students allowed us to exclude the effect of antipsychotics on the CNVs of the three repeats during different periods of life.

The major finding of the study may be formulated as follows: During a period of psychoemotional stress, the f-SatIII and TR contents significantly change in the DNA isolated from human blood leukocytes, whereas the rDNA CNs remain stable. The changes in cases and controls are not associated with a mental disease nor with a therapy, but only with psychoemotional stress itself.

### 4.1. Psychoemotional Stress Induces Oxidative Stress in the Human Body

A question arises: ‘What mechanism can account for the f-SatIII and TR variabilities in the leucocyte DNA discovered in students during different periods of the educational process?’ We have previously shown that oxidative stress induced by external factors (e.g., low doses of ionized radiation), or by endogenous stress inherent to SZ, causes considerable f-SatIII variabilities in human cells [18]. In SZ patients, oxidative stress is upregulated due to a pro- and anti-oxidative imbalance. High ROS levels are detected both in the brain and blood cells of SZ patients [57,58,59,60,61,62,63,64]. Therefore, in order to explain the variability of the two repeats in the students’ DNA, we analyzed the oxidative stress levels, which could potentially change in response to stress in students, and compared them to the levels of stress in SZ patients during a period of disease exacerbation.

In our study, we used the levels of cell membrane and mitochondrial oxidoreductase NOX4 that catalyzes the hydrogen peroxide production [50] as an oxidative stress marker. NOX4 expression was detected earlier in the brain of SZ spectrum patients [65]. Figure 3 shows that the NOX4 level in lymphocytes isolated from SZ patients was significantly higher than in controls. NOX4 levels increased in the lymphocytes of students under stress. After a vacation, the NOX4 protein expression decreased significantly (Figure 3 and Figure 4).

The 8-oxodG content is a common marker that reflects DNA oxidation under ROS [51,52]. Lymphocytes isolated from SZ patients were shown to contain more 8-oxodG than lymphocytes isolated from HC. During the exam period, we found a significant 8-oxodG increase in the students’ lymphocytes up to a level comparable to that in lymphocytes of SZ patients. After a vacation, 8-oxodG decreased considerably in the lymphocytes of all the students. Stress in students was accompanied by a rise in the γ-H2AX protein, which is a marker of double-stranded DNA breaks. γ-H2AX also decreased after their vacations.

Thus, healthy medical students preparing for, and passing, their exams undergo oxidative stress comparable to that experienced by SZ patients during disease exacerbation.

### 4.2. Cellular Response to Oxidative Stress in Students

The length of telomeres (TR content) is known to diminish with aging [28,29,30]. The TR count was reduced in the DNA of SZ patients compared to healthy controls [31,32] (Figure 2). Oxidative stress may also reduce the telomere length in human cells [34]. In the present study, we found an interesting fact: during the stress period, the TR content in the students’ leucocyte DNA decreased, but then it was restored after a vacation. Human telomeres have their maximum length in young cells. Aging is always accompanied by telomere shortening. The enzyme telomerase, which is able to restore the telomere length, is inactive in normal differentiated human cells. Oxidative stress can reduce the telomere length. Thus, after stress, we could only expect a decrease in TR content in the students’ DNA, not an increase. The scheme shown in Figure 6 helps to explain the observed changes in the contents of the two repeats in the students’ DNA.

Let us consider an ideal system: a population of young cells with low levels of ROS (Figure 6). The f-SatIII content is small, while the TR content is high. The cells derived from children, and cultured cells of early passages [15], best match this ideal model. Apparently, this model can also describe the cells of healthy 19–20-year-old students during the absence of stress (St(III) subgroup).

ROS induce a f-SatIII increase (Figure 6A, brown line). The f-SatIII content grows in HSFs with replication aging, whereas the TR count decreases. Low doses of genotoxic agents also stimulate an increase in f-SatIII content [15,18]. The mechanism leading to the f-SatIII increase is still poorly understood. The most plausible version seems to be incorporation of DNA(SAT)–RNA(SAT) hybrid molecules into the 1q12 region. The hybrids are formed because of the transcription of satellite DNA in response to stress [27]. The amount of RNA(SAT3) non-linearly depends on the intensity of exposure. The cells with a high content of f-SatIII die under chronic stress and are eliminated from the cell pool [18]. In response to stress, a decrease in the TR (blue line, Figure 6A) may occur [34,35]. Stress exacerbation is correlated with a rise in cell death (black line).

Stress leads to heterogeneity between cells by the contents of f-SatIII and TR (Figure 6B). An assay of DNA isolated from the experimental cells showed an increase in the f-SatIII content and a decrease in the TR content, as compared to the “ideal” population (Figure 6C). The stress-induced death of cells contains a high content of f-SatIII and a low content of TR. With the normal elimination of defective cells from the bloodstream and a reduction in stress, a low level of f-SatIII and a high level of TR will be detected in the total DNA samples of these cells. This correlates with a decrease in the level of cells with damaged DNA (Figure 4B).

### 4.3. Cellular Response to Oxidative Stress in SZ Patients

In HC, we found an elevation in the abundance of f-SatIII and TR at average levels of the stress markers NOX4 and 8-oxodG (Figure 5C). Under moderate stress, satellite DNA transcription reached the peak values. We had previously shown that about 40% of leucocyte RNA samples, isolated from stressed healthy controls, were formed by RNA(SAT3) transcripts [18]. The satellite DNA transcription usually occurs in stressed cells or in senescent cells characterized by shortened telomeres and a disturbed heterochromatin–euchromatin balance [66]. Therefore, the upregulated satellite transcription, in parallel with the elimination of dying cells with low TR contents, are the mechanisms that increase the f-SatIII and TR contents in the cell pool.

In SZ patients, oxidative stress is significantly stronger and more chronic compared to the controls. Apparently, it leads to a decrease in the TR content of the patient’s cells (Figure 3 and Figure 5C). Genomes of SZ patients are characterized by higher rDNA CNs. The high rDNA CNs in the genome have been shown to significantly stabilize the heterochromatin shift, i.e., shifting the heterochromatin–euchromatin balance towards heterochromatin [67,68]. Probably, the low f-SatIII levels in SZ patients are associated with heterochromatin stabilization, since the satellite transcription requires a heterochromatin-to-euchromatin transition [66]. The low f-SatIII content may also be associated with the elimination of cells with high f-SatIII content from the population [18]. We have previously shown that the cellular death process was significantly activated in two-thirds of SZ patients [69]. Finally, the low f-SatIII DNA level may be accounted for by very strong stress, which blocks the satellite DNA transcription, unlike moderate stress that stimulates it.

### 4.4. Effect of rDNA Repeat Abundance on f-SatIII and TR CNVs

The ribosomal genes play a pivotal role in the cell’s life. The number of rDNA copies determines the level of ribosome biogenesis and, thus, modulates the cell’s response to stress [55]. In two previous studies and in the present one, we have reported an increase in the number of rDNA copies in the genomes of SZ patients. The causes of this remain hypothetical.

Figure 5D presents a scheme reflecting an association between rDNA copy numbers and NOX4, 8-oxodG, f-SatIII, and TR contents in SZ patients and HC. The lower rDNA CNs in the HC group is associated with a relatively high level of oxidative stress markers and increased f-SatIII and TR contents. The high rDNA CNs are associated with low levels of stress markers, low f-SatIII, and increased TR contents. Thus, in controls, larger rDNA amounts in the genome provides for a more effective cellular response to an ROS burst.

In the SZ group, low rDNA CNs are also associated with high NOX4 and 8-oxodG levels in the lymphocytes. In this subgroup of patients, NOX4 and 8-oxodG levels are much higher than in the corresponding HC subgroup. Accordingly, f-SatIII and TR contents are significantly reduced compared to the HC low rDNA subgroup. A high rDNA CN index is associated with relatively low NOX4 and 8-oxodG levels and relatively high f-SatIII and TR counts, which are comparable to the corresponding indices in the HC subgroup with a high rDNA copy number.

The high rDNA contents in the genomes of SZ patients potentially contribute to a significant reduction in oxidative stress caused by endogenous factors. This fact corroborates with the hypothesis we stated earlier, which explains the high rDNA content in the genomes of SZ patients by their embryonic selection at the stage of embryogenesis. In patients, mutations provoke genetic abnormalities. An elevated rDNA count in the embryo’s genome is only able to stabilize heterochromatin and provide for an intense ribosome biosynthesis, thus supplying the ribosome-based production of ROS-quenching enzymes in sufficient amounts, which might moderate the oxidative stress and the stress-associated DNA damage.

We believe that the further investigation of tandem repeat CNVs in the DNA samples extracted from different brain regions of mentally healthy subjects and SZ patients may provide for valuable information about the oxidative stress level and various brain regions affected by schizophrenia. Juxtaposing this information with MRI data would bring us closer to understanding the biochemical mechanisms that underpin schizophrenia.

The present and earlier studies suggest that a further investigation into tandem repeat CNVs in the human genome would provide a new perspective for understanding the mechanisms of individual resistance to psychotraumatic events. If so, medicine could be enriched with a novel technique for the prediction of individual vulnerability to psychotic diseases, including schizophrenia, posttraumatic stress disorders, neurodegenerative diseases, and longevity.

With this in mind, translational medicine should further coalesce assets of various specialists to create new insights and methods for early diagnostics of socially relevant diseases.

## 5. Conclusions

During the exam session, psychoemotional stress induced oxidative stress in medical students. The oxidative stress caused: (1) an increase in the lymphocytes’ DNA damage levels, comparable to the level of DNA damage in SZ patients during an exacerbation of the disease, and (2) significant fluctuations in f-SatIII and TR contents, whereas the ribosomal repeat contents remained stable. A hypothesis was proposed to explain the quantitative polymorphisms of SatIII and TR under transient (e.g., students’ exam sessions) and chronic (in SZ patients) stress.

## Figures and Tables

**Figure 1 genes-13-00343-f001:**
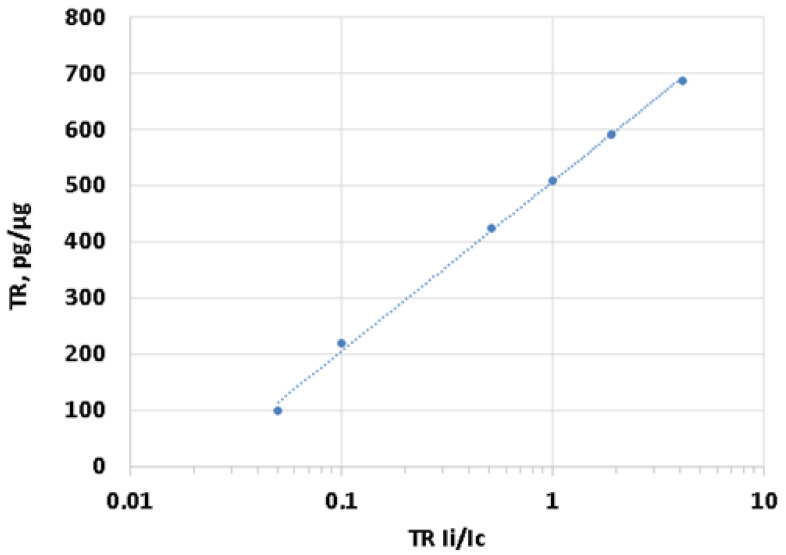
Dependence of TR content (axis of ordinates) on the relative hybridization signal (axis of abscissas) for six samples. The relative hybridization signal was calculated as a ratio of signal intensities of the inspected (Ii) and control (Ic) samples.

**Figure 2 genes-13-00343-f002:**
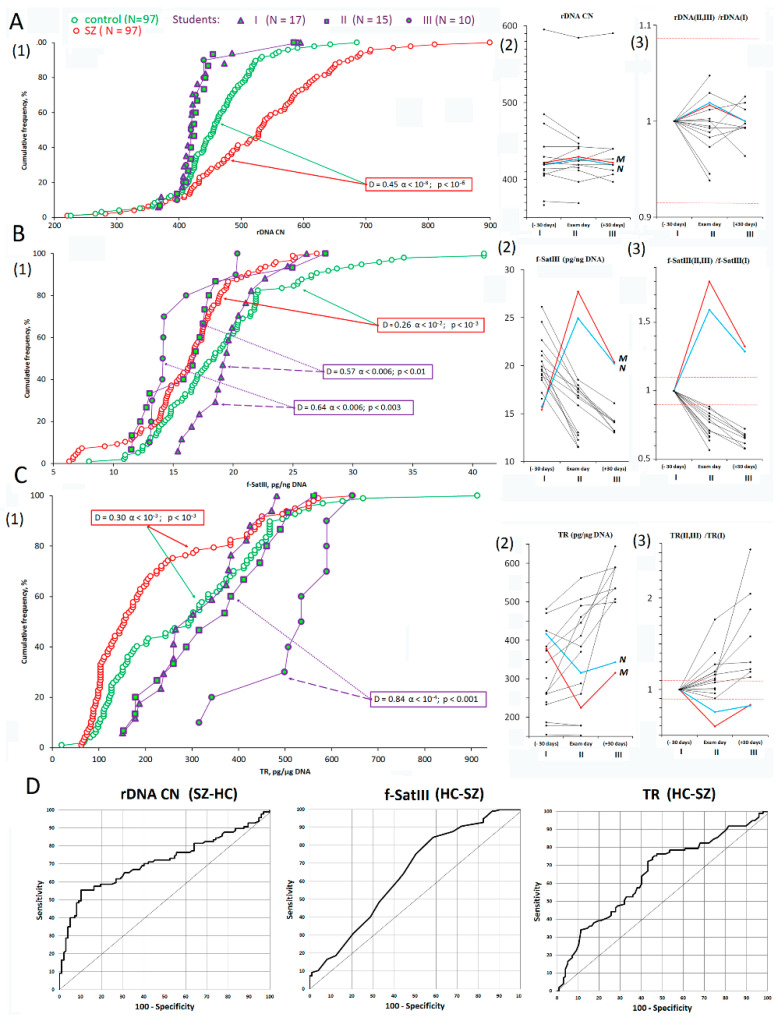
CNVs of rDNA, f-SatIII, and TR in the analyzed samples. (**A1**–**C1**). Cumulative distribution of the rDNA CN, f-SatIII, and TR in the leukocyte DNA of the SZ, HC, and St groups. See descriptive statistics for the groups in Table 2. The significance of the observed differences in the rDNA CNs was analyzed using a non-parametric Mann–Whitney U test (*p*) and Kolmogorov–Smirnov (D and α) statistics (Table 3). (**A2**–**C2**). Changes in the contents of rDNA, f-SatIII, and TR in the students’ DNA during the educational process. Straight lines connect the dots for each student. Data are highlighted for two students (M and N) who were different from the rest of the group. (**A3**–**C3**). Changes in the contents of rDNA, f-SatIII, and TR in students’ DNA during the educational process. All the data were normalized by the parameter values in the St(I) group. Red horizontal lines indicate the relative standard error (SE) of the experiment. Changes in the rDNA CN index lay within the standard error (SE), while changes in the f-SatIII and TR indices exceeded the SE. (**D**). The ROC curves plotted for the SZ and HC samples. Area under the ROC curve (AUC) is a measure of how well the parameter can distinguish between the two groups (Table 3).

**Figure 3 genes-13-00343-f003:**
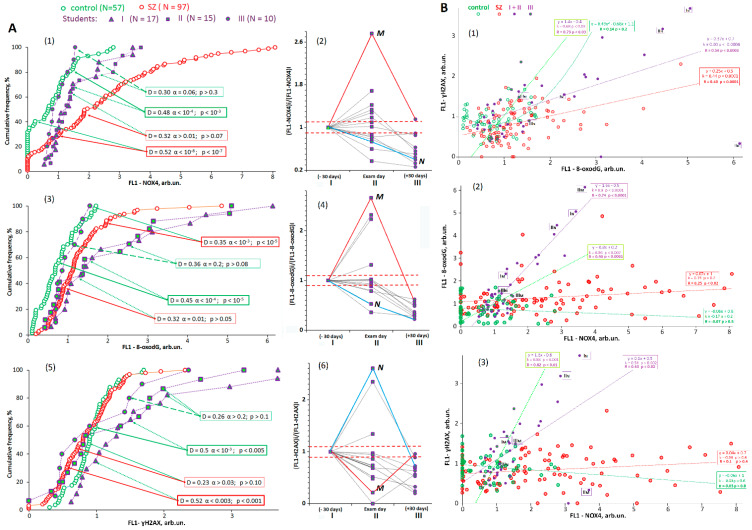
Dependence of f-SatIII content on NOX4, 8-oxodG, γH2AX, BAX levels in the lymphocytes of SZ and HC groups. (**A1**,**A3**,**A5**). Cumulative distributions of FL1-NOX4, FL1-8-oxodG, and FL1-γH2AX in the lymphocytes of SZ, HC, and St groups. See descriptive statistics for the groups in Table 4. The significance of the observed differences was tested using a non-parametric Mann–Whitney U test (p) and Kolmogorov–Smirnov (D and α) statistics (Table 3). (**A2**,**A4**,**A6**). Changes in FL1-NOX4, FL1-8-oxodG, and FL1-γH2AX during the educational process in St subgroups. All the data were normalized by the parameter values in the St(I) subgroup. Red horizontal lines indicate the relative standard error of the experiment. Changes in FL1-NOX4, FL1-8-oxodG, and FL1-γH2AX exceeded the standard error. (**B1**–**B3**). The link between FL1-NOX4, FL1-8-oxodG, and FL1-γH2AX in the samples studied. The graphs show the data for linear regression (k) and Spearman’s correlation coefficient ®.

**Figure 4 genes-13-00343-f004:**
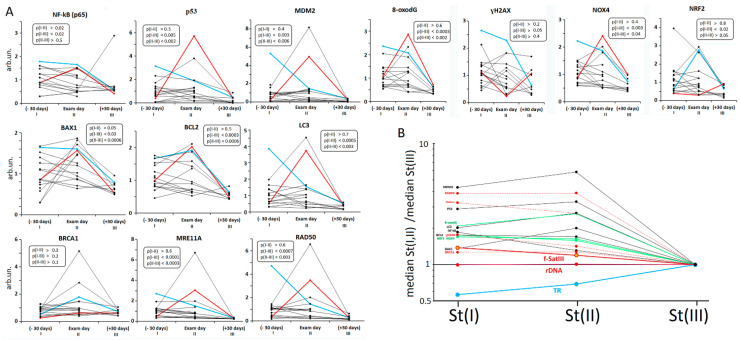
Changes in the contents of some markers in the students’ lymphocytes during the educational process. (**A**) Changes in the contents of some markers in the students’ lymphocytes during the educational process. Straight lines connect the dots for each student. The inscription on the graphs compares the three subgroups by the analyzed index (Mann–Whitney U test). (**B**) Comparison of three St subgroups by medians of the contents of the three repeats in the genome and medians of the contents of some markers in the students’ lymphocytes. All the data were normalized by the parameter values in St(III) subgroup (the period of the least stress).

**Figure 5 genes-13-00343-f005:**
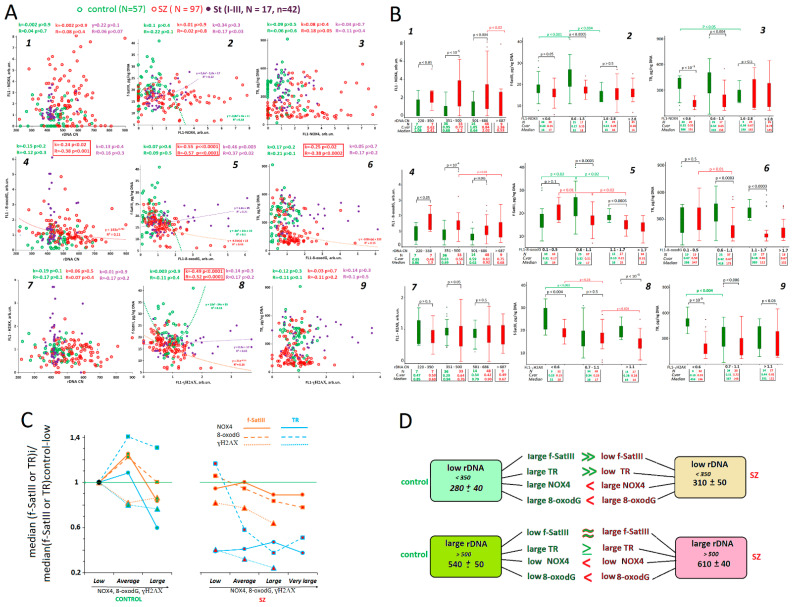
The link between the contents of rDNA, f-SatIII, TR in the genomic DNA and the contents of stress markers NOX4, 8-oxodG, and γH2AX in the lymphocytes. (**A1**,**A4**,**A7**). Dependence of FL1-NOX4, FL1-8-oxodG, and FL1-γH2AX on rDNA CNs for the samples studied. (**A2**,**A5**,**A8**). Dependence of f-SatIII on FL1-NOX4, FL1-8-oxodG, and FL1-γH2AX for the samples studied. (**A3**,**A6**,**A9**). Dependence of TR count on FL1-NOX4, FL1-8-oxodG, and FL1-γH2AX for the samples studied. The graphs show the data for linear regression (k) and the Spearman correlation (R) coefficients. (**B1**,**B4**,**B7**). Analysis of NOX4, 8-oxodG, and γH2AX in HC and SZ subgroups that differed by rDNA CNs. (**B2**,**B5**,**B8**). Analysis of f-SatIII content in HC and SZ subgroups that differed by FL1-NOX4, FL1-8-oxodG, and FL1-γH2AX levels. (**B3**,**B6**,**B9**). Analysis of TR content in HC and SZ subgroups that differed by FL1-NOX4, FL1-8-oxodG, and FL1-γH2AX. The Tables in the figures present the descriptive statistics. The graph shows sample comparisons using non-parametric Mann–Whitney U statistics. (**C**). Scheme that displays the change in the contents of f-SatIII and TR in DNA depending on the contents of NOX4, 8-oxodG, and γH2AX markers in the lymphocytes of the test groups. (**D**). Scheme showing the contents of f-SatIII and TR in DNA and the contents of the stress markers NOX4 and 8-oxodG in the lymphocytes, depending on rDNA contents in HC and SZ groups.

**Figure 6 genes-13-00343-f006:**
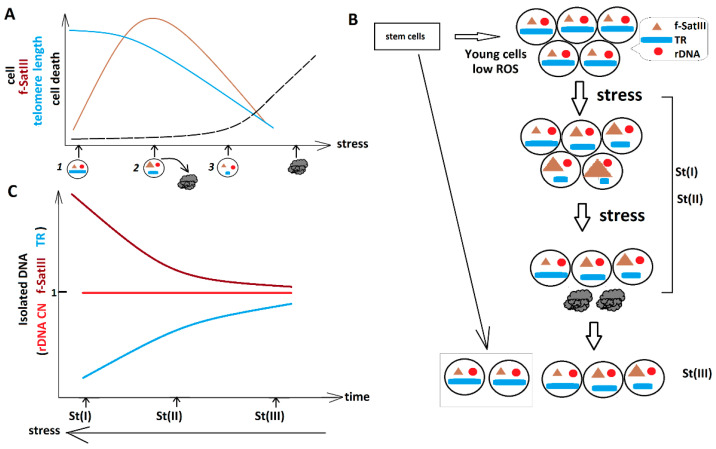
A hypothetical scheme showing the combinations of different sizes of f-SatIII and TR clusters in human cell nuclei under stress. (**A**) Changes in the contents of f-SatIII (brown line) and TR (blue line) in the cell under the stress. The stress induces cell death (black line). (**B**) Changes in the contents of f-SatIII and TR in the students’ cells. Red is for rDNA; blue for TR; brown for f-SatIII. (**C**) Changes in the contents of f-SatIII and TR in the DNA isolated from the students’ blood cells. Red is for rDNA; blue for TR; brown f-SatIII.

**Table 1 genes-13-00343-t001:** Demographic and clinical measures in schizophrenia (SZ) patients and healthy controls (HC).

Index	HC	SZ
*n*	97	97
Age	29 ± 19	27 ± 15
Age of SZ manifestation	N/A	22.1 ± 6.1
Gender (men/women)	50/47	65/32
Never smoked (%)	68	59
Smoke more than 20 cigarettes a day (%)	12	19

**Table 2 genes-13-00343-t002:** Descriptive statistics for rDNA, f-SatIII, and TR indices.

Index		SZ	HC	St(I)	St(II)	St(III)
	*n*	97	97	17	15	10
rDNA CN	Mean	531	456	430	434	437
	SD	117	72	51	47	56
	Min	221	226	367	369	397
	Max	899	686	595	585	591
	Median	529	456	419	424	421
	C*v*	0.22	0.16	0.12	0.11	0.13
f-SatIII pg/ng DNA	Mean	16	19	20	17	15
	SD	4	6	3	5	3
	Min	6	8	15	12	13
	Max	27	41	26	28	20
	Median	17	18	20	17	14
	C*v*	0.26	0.31	0.15	0.28	0.18
TR pg/µg DNA	Mean	213	291	315	349	515
	SD	149	169	103	132	108
	Min	63	20	151	153	315
	Max	646	913	482	562	644
	Median	155	298	302	370	535
	C*v*	0.70	0.58	0.33	0.38	0.21

**Table 3 genes-13-00343-t003:** The significance of the observed differences in the studied indices was analyzed using a non-parametric Mann–Whitney U test (p) and Kolmogorov–Smirnov (D and α) statistics.

		HC (*n* = 97) vs. SZ (*n* = 97)					St(I) (*n* = 17) vs. St(II) (*n* = 15)			St(I) (*n* = 17) vs. St(III) (*n* = 10)			St(II) (*n* = 15) vs. St(III) (*n* = 10)		
		K–S		U	ROC		K–S		U	K–S		U	K–S		U
		D	α	p	AUC	p	D	α	p	D	α	p	D	α	p
1	rDNA	−0.45	2 × 10^−9^	4 × 10^−7^	0.710	<0.001	−0.31	0.3	0.3	−0.15	0.9	0.7	0.17	0.9	0.6
2	f-SatIII	0.26	2 × 10^−3^	6 × 10^−4^	0.642	<0.001	0.57	0.005	0.009	0.64	0.005	0.002	0.37	0.3	0.6
3	TR	0.30	4 × 10^−4^	4 × 10^−4^	0.646	<0.001	−0.22	0.7	0.4	−0.84	8 × 10^−5^	4 × 10^−4^	−0.70	0.002	0.003
4	NOX4	−0.52	4 × 10^−9^	2 × 10^−8^	0.772	<0.001	−0.16	0.9	0.8	0.34	0.4	0.1	0.33	0.4	0.07
5	8-oxodG	−0.34	3 × 10^−4^	5 × 10^−6^	0.720	<0.001	−0.29	0.4	0.4	0.36	0.3	0.3	0.43	0.1	0.06
6	H2AX	0.23	0.03	0.2	0.567	0.149	0.29	0.4	0.3	0.40	0.2	0.09	0.23	0.8	0.4

**Table 4 genes-13-00343-t004:** Descriptive statistics for FL-NOX4, FL-8-oxodG, and FL-γH2AX indices.

Index		HC	SZ	St(I)	St(II)	St(III)
	*n*	57	97	17	15	10
FL-NOX4	Mean	0.76	2.42	1.55	1.65	1.01
	SD	0.77	2.00	0.88	0.97	0.36
	Min	0.00	0.00	0.54	0.75	0.59
	Max	2.80	8.08	3.43	3.70	1.55
	Median	0.68	1.95	1.34	1.26	0.91
	Coef. var.	1.01	0.83	0.57	0.59	0.35
FL-8-oxodG	Mean	0.75	1.26	1.91	2.13	1.00
	SD	0.47	0.76	1.51	1.64	0.32
	Min	0.10	0.19	0.42	0.57	0.69
	Max	1.70	4.85	5.05	6.13	1.55
	Median	0.68	1.12	1.40	1.45	0.89
	Coef. var.	0.62	0.60	0.79	0.77	0.32
FL-γH2AX	Mean	0.92	0.83	1.58	1.25	1.03
	SD	0.30	0.42	0.99	0.89	0.67
	Min	0.31	0.00	0.63	0.00	0.39
	Max	1.70	2.31	3.69	3.18	2.36
	Median	0.90	0.80	1.30	0.84	0.79
	Coef. var.	0.33	0.51	0.63	0.71	0.65

## Data Availability

All datasets generated for this study are included in the manuscript.

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
