# Peer review of "The Psychoemotional Stress-Induced Changes in the Abundance of SatIII (1q12) and Telomere Repeats, but Not Ribosomal DNA, in Human Leukocytes"

_genes, 2022, doi:10.3390/genes13020343_

Round 1
Reviewer 1 Report
The authors studied changes in the abundance of rDNA, SatIII (1q12) and telomere repeats in association with psychoemotional stress in schizophrenic patients and medical students. They detected a significant increase in SatIII (1q12) and decrease in telomere repeat copy number in both schizophrenics and students under stress. The main weak point of the study is in the number of analysed samples collected from the medical students that was low at the beginning of the research (N=17) and even lower in the subsequent phase (N=10).Nevertheless, the paper brings interesting new information and is acceptable after minor revision.
Comments:
-p.3, Table 1, column HC: Delete “Age of SZ onset”
-p. 4, Methods 2.3.: Add a short description of the NQH method (1 sentence)
-p. 6, l. 244-257 – This paragraph would better fit to Methods 2.1.
-p. 6, l. 259-263 – I recommend moving this paragraph to the beginning of the Discussion
-Section 3.2. – The introductory paragraphs for the individual oxidative stress markers with references would better fit to Introduction
Author Response
We highly appreciate your valuable comments. Indeed, the manuscript structure was entangled, and some text block better fitted in some other sections.
Thanks to your kind assistance, we could improve the article.
1) p.3, Table 1, column HC: Delete “Age of SZ onset” - Deleted
2) p. 4, Methods 2.3.: Add a short description of the NQH method (1 sentence) - Added, though longer than one sentence
3) p. 6, l. 244-257 – This paragraph would better fit to Methods 2.1. - Completely agree. Moved the paragraph.
4) p. 6, l. 259-263 – I recommend moving this paragraph to the beginning of the Discussion - Moved.
5) Section 3.2. – The introductory paragraphs for the individual oxidative stress markers with references would better fit to Introduction - Moved to Introduction
Reviewer 2 Report
The current study analyzed the contents of rDNA, f-SatIII and TR in the DNA of medical students and SZ patients that had undergone acute and chronic psychoemotional stress. It was concluded from the study that f-SatIII and TR contents significantly change in the DNA isolated from human blood leukocytes, whereas rDNA copy number remains stable. The study indicated that psychoemotional stress induces oxidative stress and can affect the telomere content. Overall the study is well-conducted and answered questions that were posed in the beginning.
The authors must include how this study is useful in translational medicine and how this data is valuable in clinic.
Author Response
Thank you for a good appraisal of our manuscript.
We highly appreciate your valuable comments.
As you kindly prompted, we added our ideas regarding the study usefulness in translational medicine and clinic to the end of Discussion section.